# Hummer: Towards Limited Competitive Preference Dataset

**Yusen Wu**[1][*]    **Li Jiang**[2][*]    **Junwu Xiong**[3]    **Jingqing Ruan**[3]    **Yichuan Ding**[2]
**Qingpei Guo**[3]    **Zujie Wen**[3]    **Jun Zhou**[3]    **Xiaotie Deng**[1]

[1]Peking University    [2]McGill University    [3]Ant Group

{sarinw2023, jiangli3859, junwucs}@gmail.com

## Abstract

Preference datasets are essential for incorporating human preferences into pre-trained language models, playing a key role in the success of Reinforcement Learning from Human Feedback. However, these datasets often demonstrate conflicting alignment objectives, leading to increased vulnerability to jailbreak attacks and challenges in adapting downstream tasks to prioritize specific alignment objectives without negatively impacting others. In this work, we introduce a novel statistical metric, Alignment Dimension Conflict, to quantify the degree of conflict within preference datasets. We then present Hummer and its fine-grained variant, Hummer-F, as innovative pairwise preference datasets with reduced-conflict alignment objectives. Hummer is built based on UltraFeedback and is enhanced by AI feedback from GPT-4, marking as the first preference dataset aimed at reducing the competition between alignment objectives. Furthermore, we develop reward models, HummerRM and HummerRM-F, which employ a hybrid sampling approach to balance diverse alignment objectives effectively. This sampling method positions HummerRM as an ideal model for domain-specific further fine-tuning and reducing vulnerabilities to attacks. Access the dataset via this link.

## 1 Introduction

Reinforcement Learning from Human Feedback (RLHF) exhibits great potential in integrating human preferences into large language models (LLMs) (Christiano et al., 2017; Ouyang et al., 2022; Bai et al., 2022; Touvron et al., 2023; Achiam et al., 2023). RLHF also holds great promise in real-world domains, such as robotics (Hu et al., 2023; Tian et al., 2023), healthcare (Yu et al., 2023; He et al., 2023), and autonomous driving (Chen et al., 2023; Du et al., 2023). A fundamental aspect of integrating human alignment objectives lies in the preference modeling stage, which crucially depends on a given preference dataset. This stage can be realized by constructing either explicit (Christiano et al., 2017; Ouyang et al., 2022; Touvron et al., 2023) or implicit reward models (Rafailov et al., 2023; Zhao et al., 2023a).

However, alignment objectives often present competing properties in current preference datasets (Biyik & Sadigh, 2018; Hong et al., 2022; Ganguli et al., 2022; Wu et al., 2024). Considering the Anthropic-HH dataset (Bai et al., 2022), emphasizing the alignment objective of harmlessness may cause an agent to offer only broad or overly cautious advice. This emphasis could prevent the agent from delivering impactful and precise guidance, which limits the capability of helpfulness. This competition dynamics among alignment objectives poses two significant challenges. On one side, it exacerbates the vulnerability of safety-trained LLMs to jailbreak attacks by crafting prompts to prioritize one alignment dimension over others (Wei et al., 2024). Besides, the conflict dynamics further complicate the attainment of equilibrium among all alignment objectives, particularly customizing models for downstream tasks that require promotion to specific dimensions ability without sacrificing performance in other alignment objectives, such as math reasoning (Azerbayev et al., 2023) and code generation (Guo et al., 2024a).

---

[*]Equal contribution, more junior authors listed earlier.

To mitigate the conflict of alignment dimensions in the preferences dataset, one line of research direction is to separately construct twisted alignment objectives via distinct reward or cost models to explicitly decouple human alignment objectives. Subsequently, these decoupled, learned models are synergized to provide a holistic preference signal tailored to specific alignment goals for downstream tasks. The integration methodology encompasses a variety of strategies, consisting of voting mechanisms (Wang et al., 2024a), differential weighting of models (Touvron et al., 2023; Wu et al., 2024), application of linear combinations (Jang et al., 2023; Rame et al., 2024), and the use of the Lagrangian multiplier (Dai et al., 2023) to manage trade-offs or to highlight specific principles of human values. However, those approaches inadvertently increase model complexity and computational overhead.

In this study, we redirect our focus toward the underlying cause of alignment conflict: the preference dataset itself. RLHF community has witnessed an emerging trend towards developing new preference datasets, driven by goals of enhancing quality and scale, incorporating fine-grained preference signal, and covering specific domains aligned with desired dimensions (Cui et al., 2023; Ji et al., 2024b; Wu et al., 2024; Stiennon et al., 2020; Lightman et al., 2023; Ethayarajh et al., 2022). Despite these efforts, a significant gap persists: *the lack of a preference dataset intentionally crafted to alleviate conflicts between alignment dimensions.* Such a dataset could potentially provide significant benefits for downstream applications that prioritize certain values (Zhang et al., 2024; Wang et al., 2024b) and reduce vulnerabilities to jailbreak attacks (Perez et al., 2022; Qi et al., 2023; Wei et al., 2024; He et al., 2024; Lyu et al., 2024; Cui et al., 2024). Moreover, there is currently no established statistical metric for assessing the degree of conflict among alignment dimensions within preference datasets.

In light of these observations, we first introduce Alignment Dimension Conflict (ADC), a statistical metric for quantifying the degree of conflict within preference datasets. This new criterion moves beyond the conventional metric of average performance across multiple objectives or domains typically featured on current leaderboards. We then present `Hummer`, standing as the first preference dataset to highlight limited competition among various alignment objectives. The construction of `Hummer` capitalizes on the advanced capabilities of AI feedback mechanisms, such as GPT-4 (Achiam et al., 2023), consisting of a three-stage process: preference & objective annotation, alignment objectives refination, and dataset split. We use the UltraFeedback (Cui et al., 2023) as our foundation dataset for this work and introduce a fine-grained version of `Hummer`, termed `Hummer-F`, which excludes the noisy preference dataset via the principle of reward gaps and compromises 80% of `Hummer`.

Based on `Hummer` and `Hummer-F`, we introduce a hybrid sampling strategy for training their respective reward models, HummerRM and HummerRM-F, based on the established Llama 2-7B model (Touvron et al., 2023). The hybrid sampling strategy achieves well-balanced performance across diverse limited-competition alignment objectives in `Hummer`, enhances resilience to jailbreak attacks, and supports further fine-tuning in downstream tasks. It accomplishes this by prioritizing certain alignment objectives without sacrificing performance in other dimensions. We summarize our contributions in two main folds:

1. We introduce the Alignment Dimension Conflict (ADC), a statistical metric for quantifying conflict in preference datasets. We then present `Hummer` and its refined variant, `Hummer-F`, designed as the first preference datasets to mitigate competing alignment objectives.

2. We develop a hybrid sampling strategy to train the reward model HummerRM from `Hummer`, balancing performance across alignment objectives and further limiting the conflict. HummerRM boosts defense against jailbreak attacks and enables downstream fine-tuning by focusing on key alignment dimensions without compromising others.

## 2   Related Work

**RLHF.**   RLHF has emerged as the leading strategy to integrate human preferences into language models through preference datasets, which can be fixed pre-collected or generated from agents or language models (Cheng et al., 2011; Akrour et al., 2011; Askell et al., 2021;

Xu et al., 2022; 2023). To integrate human values, RLHF generally obtains the final aligned policy through RL algorithms, such as PPO (Schulman et al., 2017), to maximize the reward through the trained reward model on preference datasets (Ramamurthy et al., 2022; Bai et al., 2022; Ouyang et al., 2022; Touvron et al., 2023). Another important branch is to directly anchor the human preferences to the final policy by constructing the implicit reward with policies through the closed-form optimal solution for the reward model (Rafailov et al., 2023; Zhao et al., 2023b; Azar et al., 2023; Wang et al., 2023; Ethayarajh et al., 2024; Zhou et al., 2023; Amini et al., 2024; Liu et al., 2023; Swamy et al., 2024). While these approaches are appealing for their computation cost and ease of implementation, their inherited offline paradigm suffers from the distributional shift and lack of online exploration (Guo et al., 2024b; Calandriello et al., 2024).

**Preference Datasets.** The RLHF community is observing a growing trend of new preference datasets from diverse perspectives to improve preference modeling. The dominant motivations for the introduction of new preference datasets are scalability, quality, and diversity (Guo et al., 2023; Cui et al., 2023; Wu et al., 2024). For example, SPA dataset (Guo et al., 2023) presents fine-grained (i.e., token or phrase level) feedback during optimization rather than holistic feedback during the training process. UltraFeedback (Cui et al., 2023) introduces a wide-source and high-quality preference dataset with four alignment dimensions, in contrast to two dimensions (helpfulness and harmlessness) (Ouyang et al., 2022). Besides, some recent preference datasets underscore a specific domain or alignment property (Stiennon et al., 2020; Lightman et al., 2023; Ethayarajh et al., 2022). However, existing preference datasets fail to mitigate the conflict between alignment dimensions. Enhancing the synergy of alignment dimensions improves resilience against jailbreak attacks and allows for further fine-tuning in downstream applications. This is achieved by prioritizing specific alignment objectives without compromising performance across other dimensions.

**Red Teaming LLMs with Further Fine-tuning.** Red teaming is designed to execute systematic tests and attacks on LLMs to expose their potential harmfulness and safety vulnerabilities (Perez et al., 2022; Achiam et al., 2023; Shi et al., 2024; Lyu et al., 2023; Kang et al., 2024). Recent work (Qi et al., 2023; Zhan et al., 2023; He et al., 2024) identifies that customizing policies with further fine-tuning on downstream tasks, even without harmful content, will lead to a degradation in resilience against jailbreak attacks for safety-alignment policy. We hypothesize that this degradation stems from an implicit emphasis on specific alignment dimensions, such as helpfulness, and the conflict among alignment dimensions present in downstream datasets, where the learned policy is either an implicit (DPO pipelines) or explicit distillation (PPO pipelines) of the reward models. In this work, we focus on the conflict of alignment dimensions and study further fine-tuning specific alignment dimensions on the preference modeling stage (reward model) to improve specific ability for customization tasks. Aligned with these findings, we show that further fine-tuning downstream models on desired alignment dimensions inevitably leads to performance degradation in conflicting dimensions, e.g., safety.

## 3 Preliminaries

RLHF typically starts with a generic pre-trained language model from supervised fine-tuning on a high-quality dataset for general proposes, such as conservation, noted as $\pi^{SFT}$, and then matches human preferences through a preference dataset. In this work, we mainly study the problem of competing alignment objectives in existing preference datasets.

**Preference Modelling.** One of the core ingredients of RLHF is to integrate human preferences into LLMs through preference datasets, formulated as $\mathcal{D}^P = \{x^k, y_w^k, y_l^k\}_{k=1}^K$, where $K$ is the number of total collected samples. The preference dataset $\mathcal{D}^P$ incorporates the human feedback through the preference to these two responses, i.e., $y_w \succ y_l \mid x$. Given the prompt $x$, $y_w$ denotes the preferred response by humans or advanced AI models compared to $y_l$. Given the preference dataset $\mathcal{D}^P$, we can then parameterize a reward model $r_\phi(x, y)$ and

optimize it through through the Bradley-Terry (BT) model (Bradley & Terry, 1952):

$$\max_{r_\phi} \mathbb{E}_{(x,y_w,y_l)\sim\mathcal{D}^P} \left[ \log \sigma \left( r_\phi \left( x, y_w \right) - r_\phi \left( x, y_l \right) \right) \right],$$  (1)

where $\sigma$ is the logistic function. In the context of LLMs, the reward model $r_\phi(x,y)$, is frequently constructed based on $\pi^{SFT}(y \mid x)$ by adding a linear layer on top of the final transformer layer, which yields a single scalar prediction representing the reward value (Ziegler et al., 2019; Ouyang et al., 2022).

**Policy Modelling.** Given the learned reward model $r_\phi(x,y)$ constructed through preference modeling and policy training dataset $\mathcal{D}^{RL} = \{x^k\}_{k=1}^K$, we can formulate the following optimization objective for $\pi_\theta$ to inherit the preference from $r_\phi(x,y)$:

$$\max_{\pi_\theta} \mathbb{E}_{x\sim\mathcal{D}^{RL},y\sim\pi_\theta(y|x)} \left[ r_\phi(x,y) \right] - \beta \mathbb{D} \left[ \pi_\theta(y \mid x) \| \pi^{SFT}(y \mid x) \right],$$

where $\beta$ moderates the divergence $\mathbb{D}$, such as KL divergence (Kullback, 1951), between $\pi_\theta$ and a soft-target policy $\pi^{SFT}$. This regularization ensures that $\pi_\theta$ avoids collapsing into a narrow set of high-reward actions, preserving a diverse and functional output distribution as supported by the learned reward model (Jaques et al., 2019; Song et al., 2023; Laidlaw et al., 2024; Wei et al., 2024). PPO is employed for policy optimization due to its proven efficiency and stability in training.

## 4 Hummer

This section begins with a formal definition of Alignment Dimension Conflict (ADC), a metric to assess the extent of conflicting alignment objectives in preference datasets (Section 4). We then introduce Hummer and its fine-grained variant, Hummer-F, which are datasets specifically crafted to emphasize the property of limited conflict among alignment objectives of preference datasets (Section 4.2).

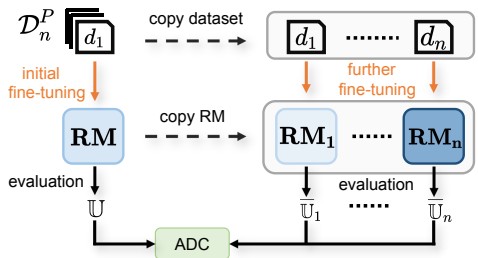

Figure 1: The ADC estimation pipeline, measuring the negative performance gap between initial and further fine-tuned reward models.

### 4.1 Alignment Dimension Conflict

$\mathcal{D}^P$ can be further organized as $\mathcal{D}_n^P = \{d_1, d_2, \cdots, d_n\}$ with $d_i = \{x^k, y_w^k, y_l^k\}_{k=1}^{K_i}$, where $d_i$ denotes the alignment dimensions, such as helpfulness in Anthropic HH dataset (Bai et al., 2022), $n$ represents the total alignment dimensions, and $K_i$ notes the total samples in dimension $d_i$ with $\sum_{i=1}^n K_i = K$. Formally, given a reward model, i.e., RM, that has been initially fine-tuned on the whole preference dataset $\mathcal{D}_n^P = \{d_1, d_2, \cdots, d_n\}$, its performance (i.e., accuracy of $\text{RM}(x, y_w) > \text{RM}(x, y_l)$) on the corresponding test dataset from $\mathcal{D}_n^P$ is represented by $\mathbb{U} = \{u_1, u_2, \cdots, u_n\}$.

To study this conflict, we copy $n$ reward models and further fine-tune each reward model on the dataset of interest for any alignment dimension, e.g., $d_i \in \mathcal{D}_n^P$, obtaining the fine-tuned performance $\overline{\mathbb{U}}_i = \{\overline{u}_{i,1}, \overline{u}_{i,2}, \cdots, \overline{u}_{i,n}\}$. The performance deviation can be obtained by $(\overline{\mathbb{U}}_i - \mathbb{U})$ of $\text{RM}_i$, where $i$ highlights further fine-tuning conducted only on $d_i$. We present the pipeline for measuring this dimension conflict, considering only negative performance deviations, in *Fig.* 1 and introduce a new statistical metric:

**Definition 1** (Alignment Dimension Conflict). *The Alignment Dimension Conflict (ADC) is defined as the second-order moment of the negative performance deviation summation on all dimensions except $d_i$:*

$$U\left[\mathcal{D}_n^P\right] \doteq \mathbb{E}_i \left[ \frac{\sum_{s\neq i}^n ((\overline{u}_{i,s} - u_s)_-)^2}{n-1} \right] \quad with \quad u_- = \min\{u, 0\},$$  (2)

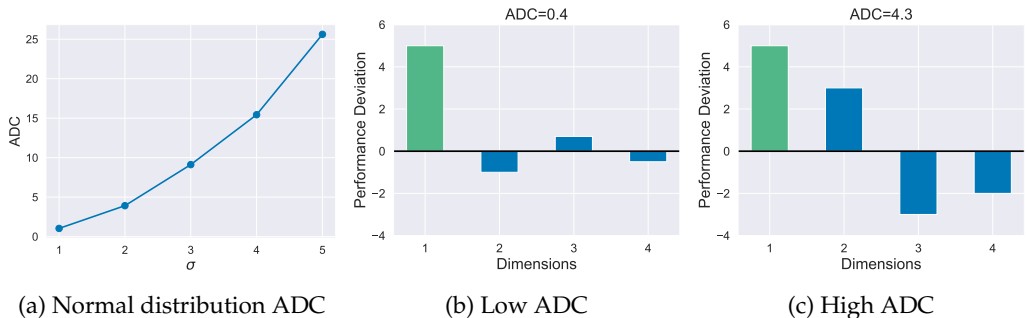

| (a) Normal distribution ADC | (b) Low ADC | (c) High ADC |

Figure 2: **(a)** Normal distribution of ADC with varying standard variance $\sigma$: $\mathbb{E}_{x \sim \mathcal{N}(0,\sigma^2)} U[x]$. **(b-c)** The performance deviation with further fine-tuning on the first dimension of preference datasets with (b) low and (c) high ADC. Intuitively, a high ADC indicates a strong conflict between the alignment dimensions of a given preference dataset.

*where $n-1$ serves as a normalization term to facilitate fair comparison for different datasets with different alignment dimensions and $\mathbb{E}_i[\cdot]$ denotes the expectation over the performance deviations obtained by further fine-tuning on alignment dimension $d_{i \in n}$ with $\mathbb{E}_i[\cdot] = \sum_{i=1}^{n} [\cdot]/n$.*

An interesting question to ask is: *What situation leads to high ADC?* We simplify the performance deviation $(\overline{\mathbb{U}} - \mathbb{U})$ sampling from a normal distribution $\mathcal{N}(0, \sigma^2)$[1]. The expression $\mathbb{E}_{x \sim \mathcal{N}(\mu=0, \sigma^2)} U[x]$ in Fig. 2a represents the ADC of a normal distribution with respect to its variance parameter $\sigma$. This measures how much adjusting one alignment dimension affects others with further fine-tuning. We observe a strongly positive correlation between ADC and $\sigma$, indicating that datasets with a higher level of competing dimensions (evidenced by greater variance on the negative side) tend to exhibit higher ADC values. The performance deviation across datasets with varying ADC levels is illustrated in Fig. 2, where datasets with low ADC are characterized by a minimal negative impact on the performance across other alignment dimensions, i.e., lower competition.

RewardBench (Lambert et al., 2024) offers toolkits for structured comparison across various properties in reward models, accommodating diverse model structures. To facilitate a systematic comparison of alignment dimension conflict levels among different reward models trained from different preference datasets, we can scale the Alignment Dimension Conflict (ADC) metric to the evaluated properties on standard evaluation toolkits, termed ADC-B, as shown in Definition 2.

## 4.2 Dataset Construction for Hummer

To decouple alignment dimensions, we introduce Hummer, the first preference dataset that aims to alleviate the competing dynamics of preference datasets. To accurately capture the multidimensionality of human preference without interference between alignment dimensions, we leverage the powerful ability of AI feedback, i.e., GPT-4, which has been heavily employed in preference dataset construction or preference modeling (Lee et al., 2023; Cui et al., 2023; Guo et al., 2023; Burns et al., 2023; Chen et al., 2024; Ji et al., 2024a). We leverage UltraFeedback (Cui et al., 2023) as the foundational dataset, attributed to its expansive scale and diversity.

We show the construction process of Hummer in Fig. 3, detailed in Appendix B. The process of identifying the limited-conflict dimension and its corresponding pairwise dataset involves three key stages: : $(a)$ **Preference annotation**: Initially, we randomly select $k = 400$ pairwise preference datasets $(x, y_1, y_2)^k$ from the foundational dataset. For each pair, we annotate preferences, alignment dimensions, and the corresponding reasons $(p, \text{dimension}, \text{reason})^k$,

---

[1]The assumption that $\mu = 0$ is justified because further fine-tuning along dimension $d_i$ might enhance performance in some dimensions while adversely competing with others.

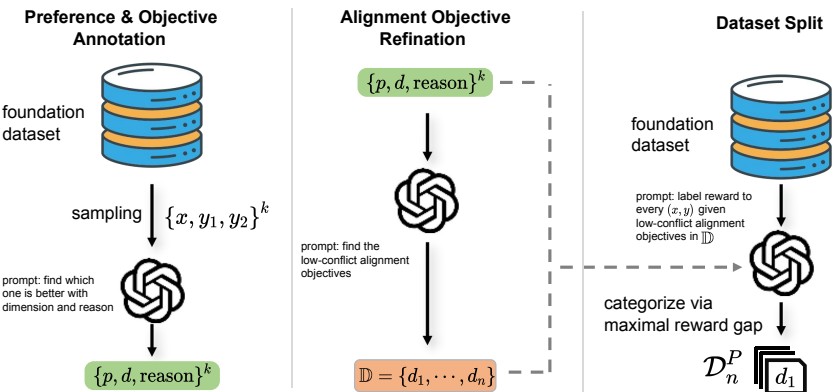

Figure 3: `Hummer` construction process. We leverage the advanced ability of GPT-4 to build `Hummer`, a preference dataset with low competitive alignment objectives.

powered by GPT-4. We totally collect 37 alignment dimensions. $(b)$ **Alignment objective refination**: We then leverage GPT-4 to refine these dimensions to minimize their conflicts and finally get $n = 6$ alignment dimensions: {'accuracy', 'conciseness', 'depth', 'empathy', 'tone', 'specificity'}. $(c)$ **Dataset split**: GPT-4 is then used to assign an absolute reward $r$ to $n$ alignment dimensions for every sample in the foundation dataset. We categorize every dataset sample $(x, y_1, y_2)$ to its corresponding dimension on the principle of maximal preference gap, i.e., $\max_{i=1}^{n} |r(x_i, y_1)_i - r(x_i, y_2)_i|$. We highlight that this splitting approach is more favorable than directly ranking as it avoids the position bias (Zhu et al., 2023) and facilitates convenience to build `Hummer-F`. `Hummer-F` is refined by applying a reward gap threshold ($\tau = 0.5$) to filter out potentially noisy preference pairs, a subset that comprises approximately 80% of `Hummer`.

## 5 Hybrid Reward Sampling

In this section, we introduce HummerRM and its variant, HummerRM-F. Both are single-reward models trained on our custom-limited competitive preference datasets, `Hummer` and `Hummer-F`, respectively. These models employ a hybrid sampling method to dynamically balance alignment dimensions, further mitigating alignment dimension conflicts in our proposed datasets, especially with imbalanced datasets over alignment dimensions, as shown in Appendix A.

Formally, considering a preference dataset with $n$ alignment objectives, denoted as $\mathcal{D}_n^P = \{d_1, d_2, \ldots, d_n\}$, we assign an initial equal sampling weight to each dimension dataset, represented by $\Lambda = \{\lambda_1, \lambda_2, \ldots, \lambda_n\}$, where $\lambda_i = 1/n$ with $i \in [1, n]$. We achieve the balance among all alignment dimensions by evaluating the preference performance across these dimensions, denoted as $\mathbb{U} = \{u_1, \ldots, u_n\}$. The sampling weights are adaptively updated in every 1 epoch (128 steps) as follows:

$$\lambda_i \leftarrow \lambda_i + \eta(\bar{u} - u_i), \quad (3)$$

where $\bar{u}$ represents the average preference performance across all alignment objectives, and $\eta$ is the temperature for updating the sampling weights $\Lambda$. To ensure adherence to the sum constraint, $\sum_{j=1}^{n} \lambda_j = 1$, we normalize the $\lambda_i$ values accordingly after every update. Consequently, the mini dataset sampled at each training step is represented by $\lfloor BatchSize \times \Lambda \rfloor$ from $D_n^p$, where $BatchSize = 128$ and $\lfloor x \rfloor$ represents the floor function.

Intuitively, if the performance of a specific dimension, e.g., $u_i$, is higher than the average ($u_i > \bar{u}$), the corresponding sampling ratio $\lambda_i$ for dataset $d_i$ decreases. Conversely, if $u_i < \bar{u}$, indicating a performance lower than the average, $\lambda_i$ increases, promoting an increasing sampling dataset for $d_i$. We then integrate all sampled datasets into one training batch and update the reward model via Eqn. (1). The hybrid sampling strategy enhances the robust performance of HummerRM across all alignment dimensions.

| Dataset | Model Type | Alignment Dimensions | Dataset Size | ADC ($\downarrow$) | ADC-B ($\downarrow$) | Reward Bench ($\uparrow$) |
|---|---|---|---|---|---|---|
| Anthropic HH | AnthropicRM | 2 | 170k | 85.04 | 204.6 | 56.72 |
| UltraFeedback | UltraRM | 4 | 64k | 67.23 | 126.3 | 68.34 |
| Hummer | HummerRM$_{\text{w/o HS}}$ | 6 | 46k | 14.35 | 38.7 | 68.55 |
| Hummer | HummerRM | 6 | 46k | 11.04 | 31.2 | 71.52 |
| Hummer-F | HummerRM-F$_{\text{w/o HS}}$ | 6 | 37k | 12.92 | 36.0 | 70.39 |
| Hummer-F | HummerRM-F | 6 | 37k | 9.62 | 28.5 | 72.13 |

Table 1: Comparison of existing preference datasets. We demonstrate that all existing preference datasets exhibit a significantly higher ADC (%) (8-10x) compared to Hummer and Hummer-F. The best performance is in  blue .

## 6   Experiments

Our testbed is designed to assess the low-conflict alignment dimensions within our introduced datasets, namely Hummer and Hummer-F. We initiate our evaluation by examining the ADC and ADC-B using HummerRM, alongside a standard reward benchmark, as detailed in Section 6.1. Subsequently, we explore the vulnerabilities of HummerRM, shown in Section 6.2. Finally, we assess the efficacy of the hybrid sampling strategy in comparison to diverse sampling methods in Appendix A.

### 6.1   Reward Model Evaluation

**Setup.** To elucidate the dynamics of low competition in Hummer and Hummer-F, we assess the ADC within their respective preference datasets; and ADC-B equipped with systematical comparison on RewardBench (Lambert et al., 2024). This evaluation is contextualized by comparisons with the Anthropic HH dataset (Bai et al., 2022), and UltraFeedback (Cui et al., 2023). Furthermore, we explore the effectiveness of hybrid sampling strategies in the training of reward models. For consistency across evaluations, we employ a consistent backbone model, specifically a fine-tuned Llama 2-7B (Touvron et al., 2023), to train the reward models for each dataset.

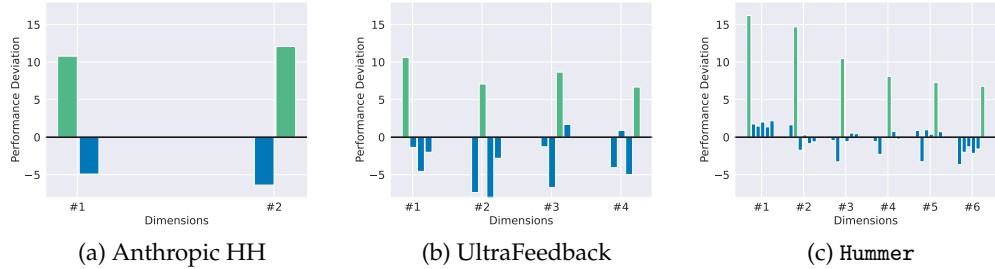

|              |                  |           |
|:---:|:---:|:---:|
| (a) Anthropic HH | (b) UltraFeedback | (c) Hummer |

Figure 4: The performance deviation with further fine-tuning on different alignment objectives, where the green bar indicates the further fine-tuning dimensions. Notably, Hummer demonstrates minimal competition among alignment dimensions.

**Result.** In Tab. 1, we summarize prevalent preference datasets with our statistical evaluation findings. Notably, Hummer and Hummer-F demonstrate a significantly reduced ADC (8-10x) compared to other preference datasets, even without hybrid sampling (HS). The structured comparison in ADC-B on RewardBench uncovers a notable consistency with the ADC results. Despite the preference dataset for Hummer and Hummer-F being considerably smaller (3-4x) than UltraFeedback, we observe an enhanced performance from HummerRM and HummerRM-F over UltraRM, by margins of 3.8% and 3.2% on RewardBench, respectively. This underscores the significance of dataset quality in preference datasets.

| Dataset | Reward model | Initial fine-tuning | Further fine-tuning on alignment dimensions of RM | | | | | |
|---------|--------------|---------------------|------|------|------|------|------|------|
| | | | # 1 | # 2 | # 3 | # 4 | # 5 | # 6 |
| Anthropic HH | AnthropicRM | 46.2 | + 6.2 | − 22.5 | - | - | - | - |
| UltraFeedback | UltraRM | 46.6 | + 4.0 | + 8.5 | + 0.3 | + 3.5 | - | - |
| Hummer | HummerRM$_{w/o\ HS}$ | 46.6 | + 3.8 | − 1.5 | + 0.5 | − 11.7 | − 2.9 | + 0.1 |
| Hummer | HummerRM | 46.4 | + 3.6 | − 1.7 | + 0.3 | − 11.7 | − 3.2 | + 0.0 |
| Hummer-F | HmmerRM-F$_{w/o\ HS}$ | 46.4 | + 2.7 | − 1.7 | + 0.8 | − 11.4 | − 3.1 | − 0.2 |
| Hummer-F | HmmerRM-F | 46.3 | + 2.4 | − 1.8 | + 0.5 | − 11.8 | − 3.4 | − 0.3 |

Table 2: Jailbreak rate $(\%, \downarrow)$ with further fine-tuning on specific alignment dimensions. While other reward models show highly fluctuating attack ratios, HummerRM demonstrates remarkable consistency with low fluctuation. Warm colors ■ are used to show increased jailbreak rates and cold colors ■ (preferred) refer to decreased jailbreak rates.

**Ablation.** The ablation study on the HS strategy reveals that improvements in ADC and ADC-B are primarily derived from our proposed datasets, while an observable margin with HS, i.e., around 3% and 7% for ADC and ADC-B respectively. Our observations confirm that HS is crucial for enhancing leaderboard-centric performance primarily aiming at "achieving a higher score" on Rewardbench. Additionally, we emphasize the importance of data quality in further fostering improvements in ADC and RewardBench. Despite these observed gains, this study fundamentally aims to identify and quantify the competing dynamics prevalent in preference datasets.

## 6.2 Jailbreak Attacks Evaluation for Reward Models

**Setup.** We posit that the HummerRM framework can mitigate vulnerabilities to jailbreak attacks by enhancing one dimension without degrading performance across other metrics. Our jailbreak evaluation framework follows Siththaranjan et al. (2023). Specifically, the jailbreak-based dataset comprises pair-wise tuples $(x, y_1, y_2)$, where $x$ represents prompts designed to elicit a harmful response from the model, $y_1$ denotes the safe response, and $y_2$ is jailbreak response (Wei et al., 2024). We quantify the 'jailbreak rate' through the proportion of instances where the reward model favors $(x, y_2)$ over $(x, y_1)$, represented by $(\sum^n \mathbb{I}(r(x, y_2) > r(x, y_1)))/n$, where $\mathbb{I}$ is the indicator function and $n$ denotes the total prompts. The higher the jailbreak rate, the greater the vulnerability of models to attacks.

**Result.** In Tab. 2, we delineate the outcomes of jailbreak attacks on Anthropic HH (Ouyang et al., 2022), UltraFeedback (Cui et al., 2023), and Hummer, with each model integrating 2, 4, and 6 alignment dimensions, respectively. Initial fine-tuning yields a uniform jailbreak rate across all datasets. Notably, UltraRM registers the highest attack rate, exhibiting a 10.4% increase following further fine-tuning on the *instruction-following* alignment dimension (# 2). This highlights a significant escalation in vulnerability to jailbreak attacks when UltraRM is specifically fine-tuned to enhance instruction-following, underscoring a pronounced tension with safety protocols. Conversely, HummerRM demonstrates exceptional robustness, with a jailbreak rate increment of less than 3% subsequent to additional fine-tuning across all dimensions. This indicates that the alignment objectives of Hummer are harmoniously integrated, ensuring that its safety remains unimpaired by further fine-tuning.

We emphasize that a declining jailbreak rate signifies enhanced defensive capabilities against jailbreak attacks. This improvement is particularly notable when further fine-tuning focuses on specific alignment dimensions, such as *harmlessness* (# 2) in the case of Anthropic HH, and *empathy* (# 4) in Hummer. Appendix Tab. 7 shows the detailed alignment dimensions for preference datasets.

**Ablation.** The ablation study on the HS indicates the strong ability of reward models against jailbreak attacks is most saturated from Hummer and Hummer-F, while hybrid sampling further enhances the defensive capabilities. These results align with those observed in the Tab. 1, affirming ADC's reliability as a proxy for quantifying preference conflicts in datasets. Addressing these conflicts is essential for maintaining resilience against jailbreaks.

| Dataset | base | Further fine-tuning on alignment dimensions of RM | | | | | | |
|---|---|---|---|---|---|---|---|---|
| | | # 1 | # 2 | # 3 | # 4 | # 5 | # 6 | Average |
| Anthropic HH | 12.1 | + 3.0 | − 1.3 | - | - | - | - | + 1.7 |
| UltraFeedback | 12.4 | + 1.3 | + 1.5 | + 1.6 | + 1.2 | - | - | + 1.4 |
| Hummer | 12.0 | + 0.8 | + 0.4 | +0 | − 0.6 | + 1.2 | + 0.3 | + 0.4 |

Table 3: Harmfulness Rate (%,↓) of policies with further fine-tuning on specific alignment dimensions of reward models, where all policies are obtained through further fine-tuning on Dolly datasets to simulate customizing fine-tuning. Hummer achieves the lowest increment of the harmfulness rate. Warm colors ■ are used to show increased harmfulness rate and cold colors ■ (preferred) refer to decreased harmfulness rate.

## 6.3 Jailbreak Attacks Evaluation for Policy Models

**Setup.** Qi et al. (2023) implies that custom fine-tuning on the policy models (even without harmlessness datasets) will significantly undermine the initial safety alignment. We can observe that the harmfulness rate increases near 12% with further fine-tuning of policies on Dolly (Conover et al., 2023) to simulate the downstream tasks, as shown in Tab. 3 (base). The harmfulness rate metric follows Qi et al. (2023) by using the GPT-4 judge which outputs a harmfulness score in the range of 1 to 5. The harmfulness rate indicates the fraction of outputs from the policy with the highest harmfulness score 5 on the test dataset. We hypothesize that further fine-tuning of reward models to highlight specific alignment abilities (e.g., helpfulness) serves as an amplifier to diminish the safety of initial alignment policies due to conflict of alignment dimensions.

**Results.** We demonstrate the results in Tab. 3 by further fine-tuning the reward models on specific alignments to prioritize certain values (such as safety in finance scenarios), which retains the same training setting in Section 6.2 but evaluates on policy levels. We can observe a subtle increment (+0.4%) on average in the harmfulness rate of policies whose reward models trained from Hummer, which implies that further fine-tuning specific alignment dimensions on Hummer does not exaggerate the loss of safety. On the contrary, the increment of the harmfulness rate of policies from Anthropic HH and UltraFeedback are 3-4x to Hummer, demonstrating an unneglectable amplification of the safety loss of policies.

## 7 Conclusion

In this study, we delve into the dynamics of competing preferences within the Reinforcement Learning from Human Feedback (RLHF) framework. We introduce a novel statistical metric termed Alignment Dimension Conflict (ADC) to quantify the extent of conflict among alignment objectives within preference datasets. We unveil the first preference dataset, Hummer, alongside its fine-grained variant, Hummer-F. These datasets are designed to mitigate dimension conflicts, facilitating domain-specific fine-tuning while increasing resilience against jailbreak attacks. This is achieved by selectively prioritizing certain alignment objectives without compromising performance across other alignment objectives. Subsequently, we develop reward models for our datasets, namely HummerRM and HummerRM-F, employing a hybrid sampling technique that dynamically adjusts the sampling weight based on reward performance across different alignment dimensions. Looking ahead, an intriguing avenue for future research lies in constructing low-conflict alignment objectives using unsupervised or self-supervised (Zhang et al., 2020; Yan et al., 2021) learning methods to discern semantic nuances. Furthermore, exploring the conflict of alignment dimensions in the preference modeling stage offers a promising avenue for understanding the safety trade-offs in further fine-tuning policies (Qi et al., 2023).

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

## A  Analysis

**Hybrid sampling strategy maintains performance on imbalanced datasets.** An imbalanced dataset arises with a non-uniform distribution of classes, often characterized by a disproportionate number of instances between major and minor classes, resulting in biased predictions (Krawczyk, 2016; Jiang et al., 2023). To investigate the efficacy of a hybrid sampling strategy in addressing dataset imbalance in the context of alignment objectives, we integrate our datasets across six alignment dimensions with a distribution ratio of $10 : 10 : 10 : 10 : 1 : 1$, where the $1 : 1$ ratio pertains specifically to specificity and tone. The results are illustrated in Fig. 5.

Fine-tuning on specific dimensions will boost the performance on its corresponding alignment dimensions but fail to achieve desirable performance on other alignment dimensions, such as Single # 1(*Accuracy*), and Single # 5 (*Tone*). We demonstrate that the All Dimensions Equal strategy, with a uniform distribution ratio of 1:1:1:1:1:1, under-performs relative to our hybrid sampling approach across all dimensions,

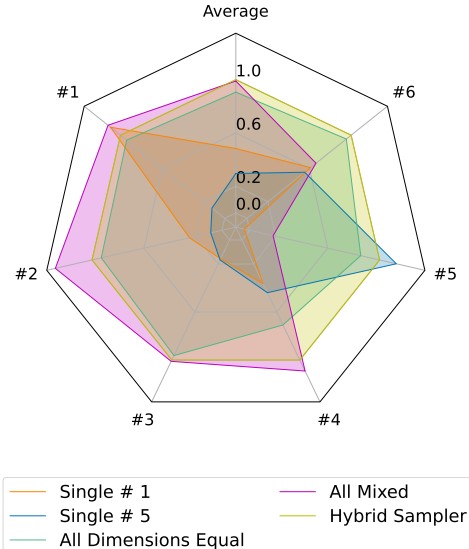

Figure 5: Performance with different sampling strategies on imbalanced datasets.

achieving only 70% to 95% of the performance of the Hybrid sampler. This implies that this uniform sampling strategy, also employed by Cui et al. (2023), may fall short in imbalanced datasets. The All Mixed strategy, integrating all alignment datasets ignoring the data balance, exhibits significantly superior performance in well-represented alignment datasets # 1 and # 2 (Depth and Accuracy), yet fails in alignment objectives with limited datasets: # 5 and # 6 (Tone and Specificity). Such an approach could further diminish the performance of lesser-represented alignment objectives, particularly in scenarios involving competing alignment objectives.

## B  Hummer **Details**

### B.1  Data Construction Prompt and Annotation

In this section, we detail the construction process of Hummer, starting from the initial data formulation. Utilizing the original dataset, we format it in the pattern $\{x, y_1, y_2, y_3, y_4\}$, where $x$ serves as the prompt and each $y_i$ represents a candidate generated by the model. To create a rich dataset for pairwise comparison, we pair the candidates, resulting in a new set of sample pairs $\{x, y_1, y_2\}$.

Following this, we select a subset of 400 pairs from this collection through random sampling. These selected pairs are then formatted into standard prompts, structured to be fed into GPT-4 for evaluation. In executing these queries, our objective is to discern the superiority between $y_1$ and $y_2$ within each pair, focusing on identifying which candidate better aligns with a specific predefined objective. Additionally, for each comparison, we aim to gather a concise explanation highlighting why one candidate is favored over the other, based on the alignment with the mentioned objective. Through this meticulous process, we identified a diverse set of 37 different objective names.

$$\{'accuracy', 'conciseness', 'depth', 'empathy', 'tone', 'specificity'\} \tag{4}$$

Subsequently, we integrate the previously identified 400 superior alignment objectives, replete with their concise explanations, into the new prompt design for GPT-4 as part of our second approach in prompt engineering. This step instructs GPT-4 to assimilate the

> **Prompt for identifying multiple objectives and definitions to reduce competing.**
>
> Following is a pair-wise RM training data item with the structure {'prompt':[prompt], 'candidate-1':[candidate-1], 'candidate-2':[candidate-2]}.
>
> ----
>
> The 'prompt' stands for a question/situation in which one agent is asked to answer; the 'candidate-1' and 'candidate-2' are two responses from agents. One response is better than the other.
> Your task is to give a brief assessment about which response is better and in which quality it did so. Your output should have following json format: {'quality':[summarize the quality name],'reason':response-1(or response-2) is better because [reason],'chosen':[0 for response-1 better and 1 for response-2 better]}. Remind the 'reason' part should contain no more than 40 words.
> Here is the item case:

> **Prompt for refining independent dimensions definitions and approaches from summarized alignment Features.**
>
> You will receive a series of example entries formatted to: {"quality": "aspect-name", "reason": "Response-1 (or Response-2) is better because [reason]}".
>
> ----
>
> Please understand the meaning of each entry in conjunction with the 'quality' and analyze the differences and connections between them.
> Finally, summarize all the 'qualities' and refine them by only retaining the 'qualities' that are semantically independent and have as little feature overlap as possible, and provide the reasons for doing so. Your output should follow this format: {"single-quality": "aspect-name", "reason": "because [reason]"}.
> Here is the list of example entries:

given information and differentiate between objectives, combining similar ones to eliminate redundancies, and then distill these into a defined set of distinct objectives. The anticipated outcome is a final set of consolidated objective names and corresponding definitions.

The sampling strategy employed in the aforementioned stages functions as a heuristic aid, steering us towards dimensionality where conflicts are minimized. Empirically, this selective approach enabled us to pinpoint ten distinct dimensions.

In the concluding procedure, we categorize the entirety of the dataset into these ten alignment objectives following the structure specified by the third prompt example. Our initial method used a singular query to present all objectives' definitions to GPT-4, subsequently prompting it to discern the most suitable alignment objective for each data entry. Unfortunately, this methodology yielded suboptimal performance due to positional bias, where objectives presented earlier were disproportionately selected over subsequent ones. The variability of results with different objective orders further indicated a lack of stability in this initial approach.

To address the limitations observed with the initial approach, we transition to a two-stage reward-ranking classification methodology. In the first stage, we present each alignment objective distinctly, pairing them with the samples for evaluation by GPT-4. Our request for GPT-4 includes assessing and assigning a reward to both $y_1$ and $y_2$ based on how well they meet the given objectives and calculating the difference between these rewards, termed the 'reward gap'. Subsequently, we compile a list of these reward gaps for each sample across the various objectives and rank them in order of magnitude. The logic underpinning this sorting

> **Prompt for final dataset splitting with objectives.**
>
> Following is a pair-wise RM training data item with the structure 'prompt':[prompt], 'chosen' :[chosen output], 'rejected':[rejected output].
>
> ---
>
> The 'prompt' stands for a question one agent is asked to answer and the 'chosen' and 'rejected' are two responses from the above agent. Your task is to assess both of them and give reward (float, 5.0 for best and 0.0 for worst) in the dimension of Depth with the definition "the thoroughness of analysis or explanation, providing detailed insights into a subject", for 'chosen' and 'rejected' responses(Each response one score). Then compute the gap between the two rewards ('chosen' reward - 'rejected' reward). Finally only output the reward gap.
> Here is the item case:

Table 4: Frequencies of Samples Aligned to Alignment Objectives under 2-stage Classification Method.

| ID | 1 | 2 | 3 | 4 | 5 | 6 |
|---|---|---|---|---|---|---|
| Dimension | accuracy | conciseness | depth | empathy | tone | specificity |
| Frequency | 15523 | 5078 | 9406 | 4011 | 2865 | 9317 |

is straightforward: a larger reward gap signifies a clear preference for one candidate over the other, primarily grounded in the specific objective, thereby determining the ultimate classification for data segregation. This iterative refinement led to the crystallization of 6 distinct alignment objectives, each defined succinctly and accompanied by the frequency of dataset samples correlating with them.

An intriguing observation emerged during this process: a notable fraction of samples (11.2%, to be precise) displayed nearly identical or very closely matched reward gaps for two or more objectives. Our strategy to address these ambiguities varies depending on the dataset context. For the standard dataset, these samples are randomly allocated to one of the objectives sharing the highest reward gap, aiming to preserve the integrity and balance of the dataset. Conversely, in the fine-grained dataset, we opt for exclusion, removing these samples outright to maintain the precision and reliability of our objective classifications.

1. **Accuracy** refers to the adherence to factual correctness, ensuring that information is free from errors.

2. **Conciseness** refers to the ability to convey information with brevity, using a minimal number of words without sacrificing clarity.

3. **Depth** refers to the thoroughness of analysis or explanation, providing detailed insights into a subject.

4. **Empathy** refers to the capacity to understand and share the feelings of others, reflecting compassion in communication.

5. **Tone** refers to the author's attitude or mood conveyed through language, influencing the reader's perception.

6. **Specificity** refers to the provision of precise and detailed information, avoiding generalizations or vagueness.

B.2    Case Study

Following Tab. 5 and Tab. 6 we provide some representative cases from `Hummer` dataset, each for one alignment objective, ranging from three lines to tens of lines. The format follows "prompt", "chosen" and "rejected" (pair-wise) versions.

### B.3 Hummer-F

While the prevailing belief advocates for the benefits of increasing dataset size to enhance the performance of language models, we emphasize the critical role of data quality over quantity in developing preference datasets for effective preference modeling. Enlarging the dataset may inadvertently incorporate noisy preference pairs, potentially diluting the integration of human values into the reward model (Siththaranjan et al., 2023; Wang et al., 2024a). In response, Wang et al. (2024a) undertook comprehensive experiments that underscored this phenomenon, proposing label smoothing and additive margin as algorithmic innovations to refine the preference model.

In light of these findings, our approach in developing Hummer involves a meticulous two-stage filtering process, resulting in the creation of Hummer-F, a fine-grained variant distilled

Table 5: Representative Cases for Depth, Tone and Specificity Objectives in Hummer.

| Objective | Number of Samples | Case of objectives |
|---|---|---|
| Depth | 2855 | "**prompt**": "What are the key factors that influence the stock market?", "**chosen**":"First, economic indicators, such as GDP growth and inflation rates, set the stage for market expectations by. reflecting the overall health of the economy. Second, corporate earnings reports can significantly impact stock prices, as they provide insights into the profitability and future prospects of individual companies. Last, geopolitical events, including trade wars and policy changes, can create uncertainty and affect investor sentiment, leading to market volatility.", "**rejected**":"The stock market is affected by economic data, company profits, and world events, but the specific details of these factors are not elaborated upon." |
| Tone | 872 | "**prompt**":"What are the potential consequences of not recycling? "**chosen**":"Not recycling can lead to increased waste in landfills, which contributes to environmental pollution and the depletion of natural resources. It's important that we consider the long-term effects on our planet and strive to reduce, reuse, and recycle whenever possible. "**rejected**": "If you don't recycle, you're practically destroying the Earth! It's people like you who are causing irreversible damage to our environment. Wake up and start doing your part, or else!" |
| Specificity | 2635 | "**prompt**": "I'm trying to improve my fitness and lose weight. What kind of exercise routine should I follow?", "**chosen**": "For a targeted approach to fitness and weight loss, commit to a routine that includes jogging or brisk walking for 20 minutes every morning, three times a week, and swimming or cycling for 30 minutes on the alternate days. On weekends, dedicate 30 minutes to strength training, focusing on full-body workouts with exercises like squats, lunges, push-ups, and planks. ", "**rejected**":"To enhance your fitness and promote weight loss, consider integrating a mix of cardio and strength training into your weekly schedule. Aim to stay active for at least 30 minutes a day, five days a week, and include healthy eating habits as part of your lifestyle." |

Table 6: Representative Cases for Accuracy, Conciseness and Empathy objectives in `Hummer`.

| Objective | Number of Samples | Case of objectives |
|---|---|---|
| Accuracy | 4721 | "**prompt**": "What is the capital of France?
"**chosen**": "The capital of France is Paris.
"**rejected**": "The capital of France is Lyon." |
| Conciseness | 1544 | "**prompt**":
"What is the primary function of
photosynthesis in plants?
"**chosen**":
"Photosynthesis converts sunlight, water,
and carbon dioxide into glucose and oxygen,
providing energy for plant growth.
"**rejected**":
"The quintessential process of photosynthesis,
a veritable symphony of nature's alchemy,
transmutes the radiant energy of the sun,
the life-sustaining elixir of water, and the
invisible breath of carbon dioxide into the
sweet nectar of glucose, the very essence
of verdant vitality, and the ethereal gift of oxygen,
all in a dance of molecular elegance that is
the cornerstone of botanical sustenance
and the tapestry of life itself." |
| Empathy | 1376 | "**prompt**":
"Ï'm feeling really overwhelmed with my work
and personal life.I'm struggling to find a balance.
What advice would you have for me?
"**chosen**":
"Ï understand that you're going through a tough time,
and it's completely natural to feel overwhelmed.
It's hard when you're juggling many responsibilities,
and it can be tough to find that balance. May I suggest
taking a few moments each day to focus on your breath
and practice mindfulness? This can help create
a sense of calm and provide a fresh perspective.
Also, remember that it's okay to say
no sometimes and protect your personal time.
"**rejected**":
"You're not alone in feeling overwhelmed.
It's a common problem in today's fast-paced world.
Here are some practical tips: prioritize tasks,
set boundaries, and consider delegating or
asking for help when necessary. Remember, it's
important to take breaks and engage in activities
that you enjoy to recharge." |

from approximately 46% of the original dataset. The initial stage utilizes scores from raw paired data sourced from UltraFeedback (Cui et al., 2023), implementing a threshold ($\tau_1 = 4.0$) on the summed score gap for initial data cleansing. This procedure effectively reduces the dataset from $N_0 = 100k$ preference pairs to $N_1 = 46k$. Subsequently, we introduce a second threshold ($\tau_2 = 0.5$) specifically within the pairwise preference datasets of `Hummer`, aiming to isolate and remove potentially noisy data based on reward signals derived from the concluding phase of `Hummer`'s assembly. This strategy further refines the dataset to $N_2 = 37k$ preference pairs. Our experimental results affirm that this meticulous dataset curation markedly enhances testing accuracy. Although the current filtering process relies on heuristic methods, future iterations could benefit from an implementation grounded in a reward modeling approach.

Table 7: The details alignment dimensions for preference datasets.

| Datasets | # 1 | # 2 | # 3 | # 4 | # 5 | # 6 |
|---|---|---|---|---|---|---|
| Anthropic HH | helpfulness | harmlessness | - | - | - | - |
| UltraFeedback | helpfulness | instruction-following | honesty | truthfulness | - | - |
| Hummer | accuracy | conciseness | depth | empathy | tone | specificity |

## C  Experiments Details

This section delineates the experimental apparatus employed in our study. Our computational setup comprised a quad-cluster of NVIDIA A100 GPUs, each furnished with 100GB of memory, providing robust computational capacity. This infrastructure was driven by a software stack anchored by Python 3.8. In the realm of deep learning libraries, we harnessed the capabilities of PyTorch version 2.0.1. Allied with PyTorch, we utilized torchvision version 0.13.1+cu113 and torchaudio version 0.12.1+cu113 to manage image and audio data transformations, respectively. Additionally, scikit-learn version 1.0.1 served as our machine learning toolkit, offering a versatile assortment of algorithms for data mining and analysis.

To expedite the training process, we integrated the flashattention library at version 1.0.0, specifically optimized to harness the A100's computing prowess effectively. This library was instrumental in reducing the computing overhead significantly, thus accelerating training times for our models.

Below, we expand on the specifics of our experimental methodologies, ensuring that we shed light on each significant aspect that could possibly influence the replicability and interpretation of our research findings.

### C.1  Datasets evaluation

We initiate our experimentation by training an encompassing model on the Hummer dataset utilizing the LLaMA2-7B architecture, extending over $m_0 = 24000$ training steps. To assess its performance, we deploy the model to RewardBench, yielding an evaluative score.

To gauge and juxtapose the Average Deviation Coefficient (ADC) across varying datasets, we embark on a fine-tuning regimen. This phase commences with models that have undergone a warm-up phase of training, aligned with different specified objectives. These fine-tuned models, including the initially warmed-up model, undergo individual assessments against the corresponding evaluation sets of each dataset. The objective is to discern the adjustment in prediction accuracy specifically on the RM dataset. We normalize the observed changes to derive the relative variation and, leveraging the ADC as previously defined, calculate the precise value through the established formula.

To mitigate the potential biases introduced by the model architecture in evaluating datasets, we standardize the use of the Llama2-7B model as our foundational model for all datasets undergoing evaluation. This standardized approach includes an initial phase of training amounting to $k_0 = 1000$ steps, covering the entirety of the source dataset—a conglomerate reflecting the diverse spectrum of the target evaluation dataset. This foundational model subsequently anchors the further fine-tuning training sessions and comparative performance analyses.

In the advanced fine-tuning phase, we meticulously sample from each subset within the evaluation dataset, catering to distinct alignment objectives. This step involves engaging in reward model training over $M = 4000$ steps, rooted in the preliminarily trained base model. For those datasets facing a data scarcity, we incorporate a multi-epoch replay and reuse strategy. This method is pivotal in circumventing the undue repetition of data samples, thereby minimizing the risk of overfitting and maintaining the model's generalization capabilities.

### C.1.1 ADC calculation

For the computational demands of our experiment, each further fine-tuning phase of our model on the `Hummer` dataset, leveraging the LLaMA2-7B framework, required approximately 6 hours of dedicated processing time using four NVIDIA A100 GPUs. Our ADC evaluation involves the following three key steps:

($a$) **Single evaluation strategy**: During the evaluation stage, we strategically sample 1,000 instances from each test set corresponding to the distinct alignment objectives integrated within our target evaluation dataset. This sampling aims to rigorously assess the prediction accuracy of our fine-tuned models. Adopting a standard reward model evaluation approach, we analyze the competency of the reward model by presenting two candidate responses to a given prompt. The evaluation criteria are straightforward: if the candidate response marked as "chosen" garners a higher score compared to its counterpart across the sampled data, the model's prediction for that instance is deemed accurate; conversely, it's labeled inaccurate. The precision of the model, thus, is quantified as the percentage of instances correctly evaluated as accurate.

($b$) **Evaluate further tuning**: Upon refining the new model via further fine-tuning on an alignment objective from the base model, we meticulously evaluate the impact of this fine-tuning on relative accuracy across all objectives delineated in the dataset. In analyzing the outcomes, our focus narrows to the adverse effects—specifically, the reduction or negative impact that further tuning dedicated to one objective might have on the performance across other objectives. This analysis is operationalized by computing the squared mean of these reductions.

($c$) **Compute ADC value**: Concluding this multi-faceted evaluation, we compute the expectation across each specified objective within the target evaluation dataset. This computational step culminates in the derivation of the Average Deviation Coefficient (ADC) result, effectively encapsulating the nuanced dynamics our definition intended to capture. This ADC measurement serves as a nuanced indicator, reflecting the model's balanced performance across a spectrum of alignment objectives, shedding light on the intricate trade-offs that underlie fine-tuning processes in deep learning model optimization.

### C.1.2 ADC-B calculation

Formally, the performance of a given reward model after fine-tuning on its preference dataset $\mathcal{D}_n^p$ is denoted as $\mathbb{V} = \{v_1, v_2, \cdots, v_m\}$, where $m$ indicates the total dimensions of abilities for assessment, e.g., Reasoning ability in RewardBench. With further fine-fune of the reward model on one specific dimension $d_i \in \mathcal{D}_n^p$, new evaluated performance and benchmark performance deviation are defined as $\overline{\mathbb{V}}_i = \{\overline{v}_{i,1}, \overline{v}_{i,2}, \cdots, \overline{v}_{i,m}\}$ and $\overline{\mathbb{V}}_i - \mathbb{V}$, respectively. We then can evaluate the ADC of datasets with a structured comparison on standard benchmarks:

**Definition 2** (Alignment Dimension Conflict Benchmark). *The Alignment Dimension Conflict (ADC) extended to standard benchmark evaluation is the second-order moment of negative performance deviation on all evaluation dimensions in the benchmark:*

$$V\left[\mathcal{D}_n^p\right] \doteq \mathbb{E}_i\left[\frac{\sum_{j=1}^m ((\overline{v}_{i,j} - v_{i,j})_-)^2}{m}\right] \quad with \quad v_- = \min\{v, 0\}, \tag{5}$$

### C.2 Hybrid Sampler

To rigorously evaluate our novel hybrid sampler methodology against the conventional fixed-ratio mixture sampling technique, we undertake comparative training experiments using the same dataset. In this case, we exemplify the process with the fine-grained version of the `Hummer` dataset. We standardize the foundation of our comparative analysis by utilizing the Llama2-7B base model, maintaining a consistent training duration of $N = 2000$ steps across all experimental trials. Post-training, we assess the resulting reward model's performance on various objectives' evaluation sets within the `Hummer` dataset. The findings related to relative accuracy are illustrated in Fig. 5.

**Parameters setting**Articulating the specifics of the hybrid sampler configuration, we establish the following parameters: each objective weight, $\lambda_i$, is set at the uniform value of $1/6$, corresponding to an equal division of focus across all objectives. The adherence threshold, $threshold_i$, is set to 0.80, indicative of our criterion for sample selection consistency. Moreover, the learning rate (denoted as lr) for the $\lambda$ values is calibrated at $1e - 4$. These weights, $\lambda_i$, subsequently inform the proportional sampling across the respective datasets, such that the ideal number of samples from dataset $i$ in a single batch would approximate to $BatchSize \times \lambda_i$.

**Handling sampling size not integer**:Addressing scenarios when the calculated sampling size for specific objectives does not yield an integer, we initially resort to the floor function, expressing this as $SampleSize_j = \lfloor BatchSize \times \lambda \rfloor$. Post-computation, we then determine the remaining sampling capacity, described as $BatchSize - \sum SampleSize_j$. The ensuing step entails random sampling for the objectives that correlate with this remaining budget, relying on $\Lambda_j$ as the probability factor. This tailored approach aims to uphold the integrity of equitable consideration for each alignment objective, meticulously adhering to the preset guidance of $\Lambda_j$. Such stringent adherence seeks to ensure the sampler's fairness and objectivity across the landscape of alignment objectives within the dataset.

**Result analysis**The radar chart reveals notable findings regarding the performance of the hybrid sampling methodology within a fixed training-step regime. Specifically, the hybrid sampler's performance closely matches the precision gains seen when training objectives independently (showing a difference of less than 5.6%) for accuracy and conciseness objectives. Additionally, this approach yields a higher precision improvement rate (by roughly 4.3%) than that of the fixed-ratio 1:1 mixture sampling method for the same objectives. When juxtaposed against the equal-ratio 1:1:1:1:1:1 mixture sampling strategy spanning all six objectives of the dataset, the hybrid sampler shows an even more marked enhancement, outstripping the uniform mixture method by over 10%. Significantly, the hybrid sampling approach also surpasses strategies that forgo additional fine-tuning for the remaining four objectives. The rationale behind these outcomes can be intuitively understood when considering how objectives, which are not specifically bolstered by increased sample counts—the same FLOPs (Floating Point Operations Per Second)—can still be affected to different extents, as indicated by the dataset's ADC levels. Some objectives might lag in improvement when provided with the same or smaller sample distribution proportions. The hybrid sampler intelligently adjusts for this by diminishing the sampling proportions of objectives that have already attained a satisfactory level of accuracy enhancement. This reallocation tactic beneficially channels a greater share of the training proportion towards those objectives that show slower gains. Consequently, this method maximizes training efficiency, enabling more substantial improvements under a constant computational budget.

