# OpenReview forum: "Hummer: Towards Limited Competitive Preference Dataset"
_colmweb.org/COLM/2024/Conference — COLM_

### Official Review · Reviewer_6Tue · 2024-05-08

**Rating:** 5
**Confidence:** 2
**Ethics Flag:** 1

**Summary:**

This paper tackles the problem of balancing between different, potentially conflicting alignment objectives in preference datasets (e.g. helpfulness vs. harmlessness). The paper first introduces a metric (ADC) that measures the RM performance deviation on other datasets when fine-tuning on data for a single dataset (each dataset presumably corresponding to a different alignment objective). Then the paper proposes a sampling strategy to get a more balanced RM dataset (Hummer) w.r.t. to the different dimensions. The authors show that the new dataset achieves better ADC scores.

The main issue with the paper is that it’s not written clearly enough so that it would be easy to understand what exactly is the problem it’s trying to solve (beyond the high-level idea of balancing between conflicting alignment objectives). Some examples of unclear writing below:
* Please give concrete examples of what you mean by “alignment dimension” already in the abstract/introduction.
* page 2: Redundant sentences one after another: “Despite these advancements, there remains a notable gap: the absence of a preference dataset specifically designed to mitigate the conflict between alignment dimensions. [...]  Despite these efforts, a significant gap persists: the lack of a preference dataset intentionally crafted to alleviate conflicts between alignment dimensions.”
* page 6: “We categorize every pair of datasets” Do you mean “every pair of responses”?

Moreover, it’s unclear how useful e.g. the proposed dataset would be in practice beyond helping to optimize the ADC score proposed in the paper.

**Questions To Authors:**

Fig. 4 suggests that the different datasets use different alignment dimensions. Are the ADC values between the datasets still comparable? Why not use shared dimensions?

**Reasons To Accept:**

1. The problem of balancing between different objectives during RLHF is of great practical importance.

**Reasons To Reject:**

1. The paper is not written clearly enough to properly motivate the problem and describe the proposed solution in a way that it could be easily reproduced.

2. The practical usefulness of the proposed dataset is unclear (beyond it helping to optimize the score proposed in the paper).

---

> ### Author Rebuttal · Authors · 2024-05-28
>
> Thank you for reviewing our work! However, we respectfully believe that the essence and contributions of our work have been significantly misunderstood.
>
> **Response to Reject 1** (*paper writing*): Note that alignment dimensions $d _i=${$x^k, y _w^k, y _l^k$}$ _{k=1}^{K _{i}}$ refer to the sub-set of the preference dataset: such as "Harmless" highlighting safety in Anthropic HH. For others, please see the new [manuscript](https://anonymous.4open.science/r/COLM2024_587_hummer-75AD).
>
> **Response to Reject 2** (*The practical usefulness of Hummer*): We respectfully argue that the contributions of our work are:
> 1. We are the first to quantify the conflict in preference datasets and introduce a new statistical metric.
> 4. We introduce a new preference dataset to mitigate the conflict.
> 5. We develop a hybrid sampling strategy to train reward models to improve interested metrics.
>
> The practical usefulness of Hummer includes (even without hybrid sampling):
>
> 1. **Improved performance on RewardBench**: We observe 2% improvement on RewardBench over UltraFeedback, despite Hummer originating from UltraFeedback. (Table 1).
> 2. **Insights into dataset quality**: we emphasize the importance of data quality in further improvements in ADC and RewardBench with Hummer-F (Table 1).
> 3. **Boosts defense against jailbreak attacks**: With further fine-tuning, beyond low ADC, we observe that HummerRM exhibits strong robustness against jailbreaks (Table 2).
>
> Also, in related work, we provided a plausible explanation for safety compromises during further fine-tuning of policies in the new version: *the emphasis on certain alignment dimensions and conflicts among them ([Qi et al, ICLR (2024)](https://arxiv.org/abs/2310.03693))*. We are expanding our jailbreak experiments with further policy fine-tuning to reinforce our hypothesis. We plan to incorporate these results in a future version and, hopefully, share them by the end of the rebuttal, pending resource availability.
>
> **Q 1.1** (*Comparison of ADC*): Yes! it remains comparable due to the normalization (n-1 in the denominator) and expectation term, making it a "mean" metric independent of the dimension number.
>
> **Q 1.2** (*Why not use shared dimensions?*): Existing preference datasets and alignment dimensions are well-constructed. This work is motivated by conflicts between alignment dimensions, necessitating new dimensions and datasets. However, we recognize potential overlaps in alignment dimensions across different datasets.

---

> > ### Author Response · Authors · 2024-06-01
> > **Rebuttal by Authors (2/2)**
> >
> > We respectively agree with Reviewer 9qZi and remove the anonymous link for the paper to make fairness to other
> > submissions and limit the burden of reviewers. In the following, we provide further clarification. Here we provide further essential clarification for our prior rebuttal.
> >
> > **Response to Reject 1.3** (*paper writing: redundant sentences*): Thanks for this notification and we will make the correction in our final version.
> >
> > **Response to Reject 1.3** (*paper writing: we categorize every pair of datasets” Do you mean “every pair of responses”?*):  This refers to every dataset sample $(x, y _1, y _2)$.
> >
> > Regarding notations, here we also provide essential updates for notations in Section 4.1: $\mathcal{D}^{P}$ can be further organized as $\mathcal{D}^{P} _n=${$d _1, d _2, \cdots, d _n$} with $d _i={x^k, y _w^k,  y _l^k} _{k=1}^{K _{i}}$, where $d _i$ denotes the alignment dimensions, such as helpfulness in Anthropic HH dataset, $n$ represents the total alignment dimensions, and $K _{i}$ notes the total samples in dimension $d _i$ with $\sum _{i=1}^n K _i=K$}. Formally, given a reward model, i.e., $\text{RM}$, that has been initially fine-tuned on the whole preference dataset $\mathcal{D}^{P} _{n}=\{d _1, d _2, \cdots, d _n\}$, its performance (i.e., accuracy of $\text{RM}(x, y _w) > \text{RM}(x, y _l)$) on the corresponding test dataset from $\mathcal{D} ^{P} _{n}$  is represented by $\mathbb{U} = ${$u _1, u _2, \cdots, u _n$}.

---

> > ### Author Response · Authors · 2024-06-06
> > **Policy Jailbreak experiments on Harmful Samples**
> >
> > | Dataset       |                      | Dim1    |            | Dim2    |            | Dim3    |            | Dim4    |            | Dim5    |            | Dim6    |            |
> > |---------------|----------------------|---------|------------|---------|------------|---------|------------|---------|------------|---------|------------|---------|------------|
> > |               |                      | Initial | Fine-tuned | Initial | Fine-tuned | Initial | Fine-tuned | Initial | Fine-tuned | Initial | Fine-tuned | Initial | Fine-tuned |
> > | hh            | Harmfulness Score    | 1.08    | +3.72       | 1.03    | +1.97      | -       | -          | -       | -          | -       | -          | -       | -          |
> > |               | Harmfulness Rate     | 0.3%    | +52.1%      | 0.3%    | +39.2%     | -       | -          | -       |            | -       | -          | -       | -          |
> > | Ultrafeedback | Harmfulness Score    | 1.07    | +2.63      | 1.10    | +2.79      | 1.06    | +2.52      | 1.07    | +2.59      | -       | -          | -       | -          |
> > |               | Harmfulness Rate     | 0.3%    | +51.5%     | 0.4%    | +54.1%     | 0.3%    | +49.7%     | 0.3%    | +50.3%     | -       | -          | -       | -          |
> > | Hummer        | Harmfulness Score    | 1.07    | +2.57      | 1.05    | +2.3       | 1.06    | +2.37      | 1.03    | +2.11      | 1.06    | +2.35      | 1.06    | +2.51       |
> > |               | Harmfulness Rate     | 0.3%    | +50.8%     | 0.3%    | +44.1%     | 0.3%    | +47.2%     | 0.3%    | 43.2%      | 0.3%    | 44.3%      | 0.3%    | +49.6%     |
> >
> >
> > We provide the jailbreak experiments of policy levels on 10 harmful examples for 5 epochs, following the experiments on ([Qi et al, ICLR (2024)](https://arxiv.org/abs/2310.03693), Table 1).  Specifically, the GPT-4 judge outputs a harmfulness score in the range of 1 to 5, with higher scores indicating increased harm. The Harmfulness Score and Rate indicate the average harmfulness score and the fraction of test cases that receive the highest harmfulness score 5, respectively.
> >
> > Analysis of the data presented in the table reveals that, on average, the increment in harmfulness score for Hummer is less pronounced than for UltraFeedback. Additionally, Hummer maintains a lower increase in the Harmfulness Rate, staying below 50% in 5 out of 6, in contrast to UltraFeedback, which achieves this in only 1 out of 4.

---

> > > ### Author Response · Authors · 2024-06-06
> > > **A Gentle Reminder**
> > >
> > > Dear Reviewer,
> > >
> > > This is a gentle reminder that we have submitted the rebuttal to address your comments. We sincerely appreciate your feedback and are happy to address any additional questions you may have during this discussion period. We thank you again for taking the time to review our work.
> > >
> > > Best regards,
> > > Authors

---

### Official Review · Reviewer_gJMS · 2024-05-11

**Rating:** 6
**Confidence:** 4
**Ethics Flag:** 1

**Summary:**

In this paper, the authors introduce the concept of Alignment Dimension Conflict (ADC) as a metric to quantify conflicts within preference datasets, particularly in the context of RLHF. They present the Hummer to measure conflicting alignment objectives in preference datasets, showcasing reduced ADC compared to other datasets. The authors further train reward models, HummerRM and HummerRM-F, using a hybrid sampling approach to effectively balance diverse alignment objectives. The analysis reveals a negative correlation between ADC and the number of alignment objectives, suggesting that incorporating more fine-grained alignment metrics could help mitigate conflicts to some extent.

**Reasons To Accept:**

1. One strength of the paper lies in its detailed construction of the Hummer dataset, where pair-wise data samples are systematically generated and assessed to identify specific alignment objectives.

2. The two-stage reward-ranking classification method used in this paper allows for a comprehensive evaluation of alignment objectives, leading to the identification of key dimensions with minimal conflict.

3. Additionally, the paper provides a structured approach to refining independent dimensions and aligning them with human preference.

**Reasons To Reject:**

Regarding the weakness: this work lacks a thorough discussion on the generalization of various dimensions of human values within the dataset construction process. To improve, 1) the authors could enhance the paper by elaborating on how the alignment objectives identified in the Hummer dataset that can be applied to a wider range of situations / datasets. This might be useful for  evaluating the robustness and utility of the dataset. 2) the paper would also benefit from a more detailed test of the weak-to-strong generalization in alignment. Such an analysis could provide a clearer understanding of how well the proposed methods perform when extended beyond their original setups.

---

> ### Author Rebuttal · Authors · 2024-05-27
>
> Thank you for appreciating our work and highlighting the critical issues of generalization over various dimensions of the proposed dataset! We provide the detailed responses in the following:
>
> **Response to Reject Reason 1** (*lacks a thorough discussion on the generalization of various dimensions*): We respectfully argue that this is not feasible at the current stage due to the following reasons:
> 1. This requires a significant computational budget, and the evaluation pipeline is considerably long and time-consuming.
> 2. Prior work, such as Anthropic HH and UltraFeedback, designs the alignment dimensions through heuristics.
> 3. We acknowledge that there may be better approaches to constructing the desired limited-competitive preference dataset over dimensions through unsupervised learning or other methods.
>
> **Response to Reject Reason 1.2** (*the paper would also benefit from a more detailed test of the weak-to-strong generalization in alignment.*): We respectfully believe that there may be some misunderstandings regarding the relationship of this work to weak-to-strong generalization in alignment. Notably, weak-to-strong generalization [1] typically introduces a novel methodology for RLHF to improve the benchmark performance. Conversely, our work aims to identify and measure the competing dynamics inherent in preference datasets. However, we acknowledge the possibility of misunderstanding the question posed. If Reviewer gJMS is interested in specific experimental designs related to weak-to-strong generalization that could further enhance our work, we would be pleased to know. We are willing to undertake additional experiments within the constraints of our computational budget.
>
> [1] Burns, Collin, et al. "Weak-to-strong generalization: Eliciting strong capabilities with weak supervision." arXiv preprint arXiv:2312.09390 (2023).

---

> > ### Comment · Reviewer_gJMS · 2024-06-05
> >
> > Thanks for the response. After read other reviewer's response, I decide to maintain the score.

---

### Official Review · Reviewer_z9ad · 2024-05-11

**Rating:** 6
**Confidence:** 4
**Ethics Flag:** 1

**Summary:**

The paper studies how to handle conflicting alignment targets (e.g., helpfulness and harmlessness) in LLM alignment.
* A metric is first introduced to measure to what extent different alignment targets conflict with each other. With this metric, the authors analyzed and compared the conflicting degree of some existing alignment dataset (Anthropic HH and UltraFeedback).
* Secondly, the authors use GPT-4 to re-label selected examples from the UltraFeedback dataset, categorizing them into six new alignment objectives. The resulting new dataset is called Hummer, and a filtered version of Hummer is also proposed named Hummer-F.
* Thirdly, the authors introduced a dynamic sampling method, which dynamically decides the proportion of samples from different alignment objectives, aiming to minimize the performance gap among the alignment objectives.

Empirical results show that the newly proposed datasets (Hummer and Hummer-F) have substantially lower conflicting levels between their alignment objectives. Applying the dynamic sampling method on the new datasets yield much more balanced results in the jailbreak tests.

**Reasons To Accept:**

* LLM alignment is a highly important task in LLM development, and how to trade off between conflicting alignment objectives is a major problem faced in LLM alignment.
* A new metric is proposed to measure the conflicting degrees of alignment objectives. This metric allows the community to better understand the relations between the alignment objectives.
* New datasets and sampling methods are proposed to balance the Reward Models’ performance across different alignment objectives. Empirical studies show promising results.

**Reasons To Reject:**

* Notations and mathematical definitions can be significantly improved. The main definitions of the metric, presented in Section 4.1, is quite confusing and hard to follow. For example:
  * $D_n^p = {d_1, …, d_n}$, but it’s unclear what is n and what is each element d_1. Intuitively, readers may expect each d_i is a tuple (prompt, winning response, losing response). However, from later descriptions, it seems that each d_i is actually a set of tuples, and hence $D_n^p$ is a set of sets. Also, what is $n$ is not described in Section 4.1? Until very late it became clear that $n$ is the number of alignment objectives.
  * In Definition 1 (Alignment Dimension Conflict), it is unclear how the expectation is defined: what is the set of elements the expectation operator operates over, and what is the probability of each element? Also for the numerator  $(\bar{u}_k - u_k)^2 _{-}$, it is unclear the square operator and the minus operator which has a a higher precedence: if the minus operator has a higher precedence, then it means $(\bar{u}_k - u_k) _{-} * (\bar{u}_k - u_k) _{-}$; however, if the square operator has a higher precedence, it means $[(\bar{u}_k - u_k)^2] _{-}$.
  * When defining ADC-B, a new subscript $m4 is introduced, but it’s unclear what is the relation between $m$ and $n$: are they completely independent, or are they the same?
  * All the above unclarities hinder the reader to fully understand the newly proposed metric.
* Ablations are needed. Promising results are presented in Table 1 and 2, but it is unclear which contributes more to the improvements, the new dataset or the new dynamic sampling method. It would be great if the authors can present two results for each dataset, one with the naive sampling strategy, and one with the new dynamic sampling strategy. It would be even better to use other sampling/integration strategies (discussed in Section 1, the last paragraph in Page 1) on each dataset, and compare the performance with the newly proposed sampling strategy.

Overall, I find the paper very interesting and it studies a highly important topic in LLM alignment, but the current paper lacks clarity and should be significantly improved before being published. I may recommend a ‘conditional acceptance’, requesting the authors to substantially improve the definitions before finally accepting the paper.

---

> ### Author Rebuttal · Authors · 2024-05-28
>
> Grateful to Reviewer Z9ad for expressing interest in the paper and constructive comments. We have updated a new version of [manscript](https://anonymous.4open.science/r/COLM2024_587_hummer-75AD/) for all concerns.
>
> **Response to Reject Reason 1** (*Notations can be significantly improved*): We briefly clarify the notation one by one:
> 1. **$D _n^p=$\{$d _1, \ldots, d _n$\}**: Yes, each $d _i$ is actually a set of tuples, for the specific alignment dimension preference dataset, such as (such as Anthropic HH with 2 dimensions: Helpfulness and Harmlessness).
> 2. **the expectation operator operates in ADC**: This expectation is over alignment dimension $(d _{i\in n})$. Say, we firstly obtain $n$ negative performance deviations via $(\overline{\mathbb{U}} _i - \mathbb{U})$ for all alignment dimensions averaged by n-1 (except the further fine-tuned one), and we take the expectation of $n$ alignment dimensions to obtain a scalar metric with $\mathbb{E} _i[\cdot]=(\sum _{i=1}^n[\cdot])/n$.
> 3. **Priority of operators**: $u _{-} = \min${$u, 0$} operator has a higher precedence. Otherwise, ADC/ADC-B $\equiv 0$ as $u^2 \geq 0$.
> 4. **Relation between $m$ and $n$**: We clarify that the number of $m$ and $n$ are indepedent. $m$ refers to an external evaluation dataset that encompasses $m$ ability dimensions; while $n$ indicates the total alignment dimension in the training preference dataset. However, we note that certain dimensions in the preference dataset may significantly overlap with abilities in Benchmark.
>
> **Response to Reject Reason 2** (*Ablations study.*): We have conducted the ablation on our proposed hybrid sampling (HS) strategy. We have three main takeaways: (1) For the mitigation of inner conflict (ADC) and jailbreak, the low-conflict dataset is the main contributor, while an observable extra margin with HS. (2) For the improvement of RewardBench, HS is the primary contributor. (3) Data quality is important for further improvements. (Tabb 1,2).
>
> We also provide a plausible explanation for safety compromises with further fine-tuning on policies: the emphasis on certain alignment dimensions and conflicts among them ([Qi et al, ICLR (2024)](https://arxiv.org/abs/2310.03693)) (Section 2). We are expanding our jailbreak experiments with further policy fine-tuning to reinforce our hypothesis. We plan to incorporate these results in future versions and, hopefully, share them by the end of the rebuttal, given the constraints on computational resources.

---

> > ### Author Response · Authors · 2024-06-01
> > **Rebuttal by Authors (2/2)**
> >
> > Thanks to the reminder from Reviewer 9qZi regarding the paper update. I respectively agree with Reviewer 9qZi and remove the anonymous link for the paper to make fairness to other submissions and limit the burden of reviewers. In the following, we provide further clarification.
> >
> > **Response to Reject Reason 1** (Notations can be significantly improved): Here we provide the essential information for the notations to address your concerns in our paper Section 4.1:  $\mathcal{D}^{P}$ can be further organized as $\mathcal{D}^{P} _n=${$d _1, d _2, \cdots, d _n$} with $d _i={x^k, y _w^k,  y _l^k} _{k=1}^{K _{i}}$, where $d _i$ denotes the alignment dimensions, such as helpfulness in Anthropic HH dataset, $n$ represents the total alignment dimensions, and $K _{i}$ notes the total samples in dimension $d _i$ with $\sum _{i=1}^n K _i=K$}. Formally, given a reward model, i.e., $\text{RM}$, that has been initially fine-tuned on the whole preference dataset $\mathcal{D}^{P} _{n}=\{d _1, d _2, \cdots, d _n\}$, its performance (i.e., accuracy of $\text{RM}(x, y _w) > \text{RM}(x, y _l)$) on the corresponding test dataset from $\mathcal{D} ^{P} _{n}$  is represented by $\mathbb{U} = ${$u _1, u _2, \cdots, u _n$}.
> >
> > **Response to Reject Reason 2** (*Ablations study.*): Here we provided ablations in Tab 1.2:
> >
> > | Dataset        | Model Type                           | Alignment Dimensions | Dataset Size | ADC $(\downarrow)$ | ADC-B $(\downarrow)$ | Reward Bench $(\uparrow)$ |
> > |----------------|--------------------------------------|----------------------|--------------|--------------------|----------------------|---------------------------|
> > | Anthropic HH   | AnthropicRM                          | 2                    | 170k         | 85.04              | 204.6                | 56.72                     |
> > | UltraFeedback  | UltraRM                              | 4                    | 64k          | 67.23              | 126.3                | 68.34                     |
> > | Hummer         | HummerRM$_{\text{w/o HS}}$           | 6                    | 46k          | 14.35              | 38.7                 | 68.55                     |
> > | Hummer         | HummerRM                             | 6                    | 46k          | 11.04              | 31.2                 | 71.52                     |
> > | Hummer-F       | HummerRM-F$_{\text{w/o HS}}$         | 6                    | 37k          | 12.92              | 36.0                 | 70.39                     |
> > | Hummer-F       | HummerRM-F                           | 6                    | 37k          | `9.62`             | `28.5`               | `72.13`                   |
> >
> > **Table 1:** Comparison of existing preference datasets. We demonstrate that all existing preference datasets exhibit a significantly higher ADC $(\%)$ (8-10x) compared to Hum and Hum-F. The best performance is in `bold`.
> >
> > | Dataset       | Reward Model                     | Initial Fine-Tuning | Further Fine-Tuning |         |        |         |        |        |
> > |---------------|----------------------------------|---------------------|---------------------|---------|--------|---------|--------|--------|
> > |               |                                  |                     | #1                  | #2      | #3     | #4      | #5     | #6     |
> > | Anthropic HH  | AnthropicRM                      | 46.2                | +6.2                | -22.5   | -      | -       | -      | -      |
> > | UltraFeedback | UltraRM                          | 46.6                | +4.0                | +8.5    | +0.3   | +3.5    | -      | -      |
> > | Hummer        | HummerRM (w/o HS)                | 46.6                | +3.8                | -1.5    | +0.5   | -11.7   | -2.9   | +0.1   |
> > | Hummer        | HummerRM                         | 46.4                | +3.6                | -1.7    | +0.3   | -11.7   | -3.2   | +0.0   |
> > | Hummer-F      | HummerRM-F (w/o HS)              | 46.4                | +2.7                | -1.7    | +0.8   | -11.4   | -3.1   | -0.2   |
> > | Hummer-F      | HummerRM-F                       | 46.3                | +2.4                | -1.8    | +0.5   | -11.8   | -3.4   | -0.3   |
> >
> > **Table 2:** Jailbreak rate (% ↓) with further fine-tuning on specific alignment dimensions. While other reward models show highly fluctuating attack ratios, HummerRM demonstrates remarkable consistency with low fluctuation.

---

> > > ### Comment · Reviewer_z9ad · 2024-06-05
> > >
> > > Thanks for the responses and the additional experiments!
> > >
> > > The authors addressed most of my questions and concerns. I would keep my positive recommendation to the paper, but would not raise the score because the authors added substantially more explanations and experiments.

---

> > ### Author Response · Authors · 2024-06-06
> > **Official Response to Reviewer z9ad**
> >
> > Thank you for your insightful comments. We concur that further detailed explanations and additional experiments would significantly deepen the understanding of the contributions and implications of our work. Moreover, we acknowledge the merit of your suggestion to conduct an ablation study without hybrid sampling, which we believe will further clarify the impact of this study.
> >
> > In the following, we also provide the jailbreak experiments of policy levels on 10 harmful examples for 5 epochs, following the experiments on ([Qi et al, ICLR (2024)](https://arxiv.org/abs/2310.03693), Table 1) to further enhance our submission.
> >
> >
> > | Dataset       |                      | Dim1    |            | Dim2    |            | Dim3    |            | Dim4    |            | Dim5    |            | Dim6    |            |
> > |---------------|----------------------|---------|------------|---------|------------|---------|------------|---------|------------|---------|------------|---------|------------|
> > |               |                      | Initial | Fine-tuned | Initial | Fine-tuned | Initial | Fine-tuned | Initial | Fine-tuned | Initial | Fine-tuned | Initial | Fine-tuned |
> > | hh            | Harmfulness Score    | 1.08    | +3.72       | 1.03    | +1.97      | -       | -          | -       | -          | -       | -          | -       | -          |
> > |               | Harmfulness Rate     | 0.3%    | +52.1%      | 0.3%    | +39.2%     | -       | -          | -       |            | -       | -          | -       | -          |
> > | Ultrafeedback | Harmfulness Score    | 1.07    | +2.63      | 1.10    | +2.79      | 1.06    | +2.52      | 1.07    | +2.59      | -       | -          | -       | -          |
> > |               | Harmfulness Rate     | 0.3%    | +51.5%     | 0.4%    | +54.1%     | 0.3%    | +49.7%     | 0.3%    | +50.3%     | -       | -          | -       | -          |
> > | Hummer        | Harmfulness Score    | 1.07    | +2.57      | 1.05    | +2.3       | 1.06    | +2.37      | 1.03    | +2.11      | 1.06    | +2.35      | 1.06    | +2.51       |
> > |               | Harmfulness Rate     | 0.3%    | +50.8%     | 0.3%    | +44.1%     | 0.3%    | +47.2%     | 0.3%    | 43.2%      | 0.3%    | 44.3%      | 0.3%    | +49.6%     |
> >
> >
> > Specifically, the GPT-4 judge outputs a harmfulness score in the range of 1 to 5, with higher scores indicating increased harm. The Harmfulness Score and Rate indicate the average harmfulness score and the fraction of test cases that receive the highest harmfulness score 5, respectively.
> >
> > Analysis of the data presented in the table reveals that, on average, the increment in harmfulness score for Hummer is less pronounced than for UltraFeedback. Additionally, Hummer maintains a lower increase in the Harmfulness Rate, staying below 50% in 5 out of 6, in contrast to UltraFeedback, which achieves this in only 1 out of 4.

---

### Official Review · Reviewer_9qZi · 2024-05-11

**Rating:** 8
**Confidence:** 3
**Ethics Flag:** 1

**Summary:**

The paper makes multiple contributions. First a metric to measure conflicts alignment dimensions in preference datasets is presented (ADC/ADC-B). Second, seeded from UltraFeedback, a preference dataset with low conflicts along 6 alignment dimensions is collected and automatically annotated with GPT-4. The usefulness of the collected dataset is demonstrated in experiments. Third, they introduce a hybrid sampling strategy for balancing alignment dimensions in preference datasets when learning a Reward Model.

**Questions To Authors:**

- Can you clarify what the “u_ = min{u, 0}” operation does? Does it mean: if \bar{u}_k - u_k > 0 -> 0 else \bar{u}_k - u_k? And I am assuming this operation is applied before the square?
- Just to clarify, ADC-B can only be applied to two (or more) datasets with data on the same alignment dimension, right?
- From Figure 4 it seems that further fine-tuning on ‘tone’ increases conflicts along the dimensions. Why do you think that is? Intutively, tone should not alter the semantics, so at least accuracy, consicneness and depth should stay somewhat equal?

Typos/Minor Comments:

- The table captions should be after the table.
- Page 3: Double semi-colon “Wang et al., 2023;; Ethayarajh et al., 2024”
- “table” Seems like table references are all lower case, but Figure references are upper case. Consider aligning.
- Section 4.2: I think this section is very hard to understand without Appendix B. Maybe it would be better to not only describe things abstractly, but more concretly e.g. by naming the alignment dimension.
- Table 2: HmmerRM → HummerRM
- Page 16: I guess there should be commas between the y’s ? “{x, y1, y2.y3.y4}” → “{x, y1, y2,y3,y4}”

**Reasons To Accept:**

- The paper makes multiple, valuable contributions (detailed in the summary above).
- The ADC metric seems particularly useful to inform future data collections for preference datasets.

**Reasons To Reject:**

The paper is often quite dense, and details are only explained in the appendix. While I acknowledge that there are many contributions, still I think it would improve the paper to at least know major experiment choices/settings, e.g. which model has been used for tuning or which alignment dimensions are comprised in Hummer.

---

> ### Author Rebuttal · Authors · 2024-05-28
>
> Thank you for appreciating our contribution, especially the significance of the ADC in measuring the conflict of preference datasets. We have updated a new version of [manscript](https://anonymous.4open.science/r/COLM2024_587_hummer-75AD/) regarding all following concerns.
>
> **Response to Q1** (*clarify for $u _{-}^2$*): Yes. $u _{-} = \min$\{$u, 0$\} operator has higher precedence than the square operator. Otherwise, ADC/ADC-B $\equiv 0$ as $u^2 \geq 0$.
>
> **Response to Q2** (*ADC-B can only be applied to two (or more) datasets with data on the same alignment dimension?*): We clarify that ADC-B is comparable with different preference datasets within different alignment dimensions. The critical distinction lies in the "dataset" for evaluation $n$ further-fined reward models on **$n$ alignment dimension of the test part of preference datasets (Hummer) VS. external dataset with $m$ ability dimensions (RewardBench)**. After that, for both ADC and ADC-B, it results in $n$ average negative performance deviations. The final ADC-B metric is holistic, calculated as the average of these $n$ deviations from $n$ further fine-tuned reward models.
>
>
> **Response to Q3** (*Figure 4 with further fine-tuning*): We sincerely thank Reviewer 9qZi for pointing out this important typo in our work! The corssponding alignment dimensions in Hummer is \{'accuracy', 'conciseness', 'depth', 'empathy', 'tone', 'specificity'\}, and the 'tone' dimension in our prior draft is actually 'specificity' (#5) in our new updated version, detailed in Appendix Tab. 6. Fine-tuning on 'specificity' (#6) will exhibit higher conflict to other alignment dimensions, e.g., 'Accuracy' (#1) focusing on free-of-error, 'Empathy' related to safety. However, we observe a contradictory phenomenon: further fine-tuning on 'specificity' (#6) results in a negative performance deviation across all other dimensions; conversely, it reduces jailbreak rates by approximately 5%. We hypothesize that the datasets designed for attacks may contain a fraction that emphasizes 'specificity' (#6) and we plan to explore this in our future work.
>
>
> **Response to Typos/Minor Comments**: Thanks for raising these concerns and please see our updated [manscript](https://anonymous.4open.science/r/COLM2024_587_hummer-75AD/).

---

> > ### Comment · Reviewer_9qZi · 2024-06-01
> >
> > Thanks for clarifying these essential details of your paper.
> >
> > I respect that you want to further improve your paper. However, linking to a revision and adding major new content is violating the conference policy. The PCs notified everyone on May 26th (i.e., two days prior to you posting this rebuttal) that this is not allowed. Doing so is unfair to other submissions obeying these rules.
> >
> > Here is the most relevant passage from the notification:
> > > We do not allow to revise the submitted content during the rebuttal and discussion period, and we limit the length of rebuttals. [...] Linking to a revision or large amount of new content will go against this policy, and we ask not to do so

---

> > ### Author Response · Authors · 2024-06-01
> > **Official Comments to Reviewer 9qZi Regarding Paper Update**
> >
> > Thanks for the important notification! The main reason why we attach our new paper is to update the notations. I respectively agree with Reviewer 9qZi and remove the anonymous link for the paper to make fairness to other submissions and limit the burden of reviewers.

---

### Decision · Program_Chairs · 2024-07-10

**Decision:**

Accept

**Comment:**

As all reviewers have agreed, the paper is tackling a very important problem in language modeling: conflict among different values during alignment phase. The proposed method in this paper to alleviate this problem is sound and novel, and also shows experimentally effectiveness. While the paper lacks clarify and needs improvement on its writing and some discussions on the values themselves, given the characteristics of CoLM and its audience, I recommend the acceptance of the paper.